# A Prospective Study of the Prevalence and Predictive Risk Factors of Repeat Breeder Syndrome in Dairy Cattle in the North of Spain

**DOI:** 10.3390/ani15020266

**Published:** 2025-01-18

**Authors:** Sofía L. Villar, Carlos C. Pérez-Marín, Jacobo Álvarez, Antía Acción, Renato Barrionuevo, Juan J. Becerra, Ana I. Peña, Pedro G. Herradón, Luis A. Quintela, Uxía Yáñez

**Affiliations:** 1Unit of Reproduction and Obstetrics, Department of Animal Pathology, Faculty of Veterinary Medicine, Universidade de Santiago de Compostela (USC), 27002 Lugo, Spain; svillarfdez@gmail.com (S.L.V.); jacobo.alvarez.torres@rai.usc.es (J.Á.); antia.accion.carro@rai.usc.es (A.A.); renato.barrionuevo@rai.usc.es (R.B.); juanjose.becerra@usc.es (J.J.B.); anaipena@usc.es (A.I.P.); garcia.herradon@usc.es (P.G.H.); uxia.yanez.ramil@usc.es (U.Y.); 2Department of Animal Medicine and Surgery, Faculty of Veterinary Medicine, Universidad de Córdoba, 14014 Córdoba, Spain; pv2pemac@uco.es; 3Instituto de Biodiversidade Agraria e Desenvolvemento Rural (IBADER), USC, Lugo Campus s/n, 27002 Lugo, Spain

**Keywords:** cow, reproductive performance, postpartum pathologies, negative energy balance, artificial insemination

## Abstract

The study addresses repeat breeder syndrome (RB), a significant reproductive issue in dairy cattle, where cows fail to conceive after three or more inseminations despite appearing healthy. The research aimed to identify risk factors contributing to RB in primiparous (first-time calving) and multiparous (multiple calvings) cows on dairy farms in northern Spain. Analyzing 2370 cows, the study identified critical risk factors such as body condition loss, reproductive pathologies (e.g., dystocia, endometritis), metabolic disorders (e.g., ketosis), lameness, and mastitis. For multiparous cows, delaying the first insemination postpartum reduced the RB risk. Seasonal effects showed fewer cases when calving occurred in summer or autumn. The findings suggest that addressing these factors through improved management, health monitoring, and breeding practices can enhance reproductive efficiency, lower costs, and support animal welfare. Therefore, reducing RB prevalence contributes to sustainable dairy farming and supports societal concerns about food production’s economic and environmental impacts.

## 1. Introduction

Repeat breeder syndrome (RB) is one of the main reproductive issues that affects cattle dairy farms worldwide [1]. This syndrome includes those animals that have normal length estrous cycles (between 17 and 25 days), and palpation or ultrasound of the genital organs show no gross evidence of anatomical abnormality or existing inflammation, or persisting pathology, but they do not become pregnant after three or more artificial inseminations [2,3,4]. This disorder has been shown to be of multifactorial etiology in which several causes may converge simultaneously, making the diagnosis, treatment, and prevention at the farm level challenging.

Over the years, numerous risk factors for RB have been described, including postpartum reproductive [5,6,7] and metabolic pathologies [8,9], production level [10], farm-related parameters [11], and climate [12]. However, the results documented are still inconsistent, as they may be affected by different management practices, such as intensive or extensive production systems; the productive and/or reproductive capacities of individual animals or herds; and/or geographic or regional characteristics. Suboptimal housing conditions, such as small and uncomfortable cubicles, limited bunk space per cow, and reduced or inadequate dry matter intake increase the risk of RB, reducing the reproductive performance and hence economic efficiency [7]. Although the risk factors that have been documented to be involved in this syndrome may vary, understanding them will significantly improve control of this important issue in dairy farms.

The prevalence of RB varies widely across studies, ranging from 9% in the United Kingdom [13], to 10.1% in Sweden [3], 11% in Bangladesh [14], 12.4% in Poland [15], 14% in Japan [12], 22% in the United States [16], 23% in Australia [17] and India [18], 25% in Spain [19], 33.9% in Ethiopia [10], and 62% in Indonesia respectively [20]. These variations may be due to diverse production systems, the management techniques used, herd size, and genetic merit, as well as the influence of environmental conditions (e.g., heat stress) to which herds are subjected, among other factors like metabolic stress due to negative energy balance (NEB) during the peripartal period [4]. While the low prevalence in the United Kingdom and Sweden may reflect the benefits of advanced production systems and management practices, Spain’s relatively high prevalence highlights gaps in its current RB prevention and management strategies. Similarly, other factors such as temperature and the consequent heat stress may interfere in the difference between Spain and colder regions. Addressing these deficiencies is critical for improving reproductive performance and underscores the importance of this study in identifying and mitigating key risk factors in Spanish dairy farms.

The economic importance of this problem is significant due to the influence that reproductive performance has on milk production [13,21]. The associated subfertility generates costs related to reduced conception rates (CR), increased days open (DO), higher culling rates, reduced life-time milk production, as well as increased replacement costs and expenses associated with veterinary care and repeated inseminations [3,22,23]. In a sector like dairy production, characterized by high production costs, reproductive efficiency is essential to ensure the profitability of dairy farms [24,25].

Regarding environmental impact, such as decreased methane emissions per liter of milk produced, it has been shown that increasing productive efficiency (through improvements in management, nutrition, reproduction, and welfare) is an effective way to reduce emissions per unit of milk [26]. On the other hand, reproductive failure is the most common reason for the involuntary culling of dairy cows worldwide [27,28], and in the current context of concern about animal welfare, it is expected that society may begin to demand longer productive lifespans according to natural cycles [29]. Considering the close relationship between reproductive efficiency and dairy productivity, identifying and treating repeat breeders is particularly important, as RB leads to prolonged calving intervals, reduced conception rates, and increased culling, all of which negatively impact overall herd reproductive performance and productivity

Therefore, this prospective study has been designed to determine the risk factors involved in RB by analyzing the reproductive data collected in dairy cattle farms in northern Spain.

## 2. Materials and Methods

### 2.1. Ethics

The procedures were conducted according to the European Union Legislation (Directive 2010/63/EU) and the Spanish Regulations for the protection of animals used for scientific purposes (RD 53/2013).

### 2.2. Animals

This study involved 2370 cows (1445 multiparous and 925 primiparous) from 4 commercial farms located in Cantabria (northern Spain). The herd sizes ranged from 265 to 370 milking cows, with an average standardized production of 12,128 L over 305 days. The total mixed rations used on all farms were based on corn silage, grass silage, alfalfa, and supplemented with concentrates. In all cases, the cows were housed in free-stall barns with individual cubicles. Three of the farms had conventional milking parlors, with cows being milked 2–3 times a day, while the fourth farm used a robotic milking system, averaging 3.1 milkings per day. In all herds, weekly reproductive examinations were performed by an experienced veterinarian, along with weekly uterine health checks that included detection and management of reproductive problems, monitoring of general health status, heat synchronization programs, and recording and analysis of reproductive data during the first four weeks postpartum.

### 2.3. Data Collection

All cows that calved between March 2020 and May 2022 on the 4 farms were monitored, with the data recorded from on-farm software as shown in Table 1.

As for multiparous cows, when the animal was included in the study, a number of historical data were also collected and stored in the farm management programs (DairyPlan (GEA, Düsseldorf, Germany) for three farms and Lely Horizon (Lely, Maassluis, Netherlands) for the remaining farm) (Table 2).

### 2.4. Statistical Analysis

Data were analyzed using the SPSS statistical software (SPSS version 28.0.1.1; IBM Corporation, Armonk, NY, USA).

To evaluate the risk factors for RB, a binary logistic regression was performed, using the backward conditional method to select the most relevant variables. In all cases, *p*-values < 0.05 were significant. Two separate analyses were conducted: one for primiparous cows and another for multiparous cows. In both cases, the dependent variable was RB (yes/no). The independent variables and their corresponding classification are represented in Table 3.

Additionally, in the analysis of multiparous cows, the following independent variables were added: calving number (2, 3, >3), 305-day standardized production in the previous calving (kg), and dry period duration in the previous calving (days). Finally, contingency tables were prepared to determine the prevalence of RB based on the variables included in the models. Cases that showed missing or doubtful data were excluded for statistical analysis.

## 3. Results

The overall prevalence of RB in the population studied was 21.1% (18.6% (11.8–29.4) primiparous and 22.8% (16.3–32.1) multiparous).

### 3.1. Risk Factors in Primiparous Dairy Cows

Table 4 shows the risk factors for RB in primiparous cows. The model obtained from the logistic regression analysis was significant and presented an R^2^ of 0.17. Endometritis (OR, 12.23; *p* < 0.01), severe postpartum body condition loss (−0.75: OR, 2.2; *p* < 0.01; −1 to −1.5: OR, 8.1; *p* < 0.01), dystocia (OR, 5.2; *p* < 0.01), clinical mastitis (OR, 4.2; *p* < 0.01), ketosis (OR, 3.5; *p* < 0.01), and lameness (OR, 3.4; *p* < 0.01) were identified as risk factors for RBS in the studied population, while summer and fall calvings had a protective effect (OR, 0.4; *p* < 0.01/OR, 0.6; *p* < 0.05). Metritis did not show a statistically significant effect (*p* > 0.05). Additionally, twin births, stillbirths, retained placenta, displaced abomasum, and subclinical mastitis were not included in the final model.

### 3.2. Risk Factors in Multiparous Dairy Cows

Table 5 presents the risk factors for RB in multiparous cows. The model obtained in this case was also significant and showed an R^2^ of 0.18. Endometritis (OR, 6.2; *p* < 0.01), ketosis (OR, 4.3; *p* < 0.01), clinical mastitis (OR, 4.2; *p* < 0.01), metritis (OR, 2.6; *p* < 0.01), lameness (OR, 2.5; *p* < 0.01), severe loss of body condition after calving (−1 to −1.5: OR, 2.0; *p* < 0.05), dystocia (OR, 1.9; *p* < 0.01), and subclinical mastitis (OR, 1.9; *p* < 0.01) were risk factors for RB in the population studied, while delaying the first insemination showed a protective effect (OR, 0.99; *p* < 0.01). On the other hand, displacement of the abomasum did not show a statistically significant effect (*p* > 0.05). Furthermore, the parity number, twin births, stillbirths, retained placenta, normalized milk yield, and dry period length in the previous calving were not included in the final model.

## 4. Discussion

In the present study, the prevalence of RB was 21.1%. Several studies have linked higher parity with an increased incidence of RB [11,14,17,40], which may be due to the development of reproductive pathologies after successive calvings, resulting in uterine alterations. Advanced age may negatively impact fertility, either due to a decline in oocyte quality or the development of ovarian abnormalities and ovulation failure caused by defects in hormone production by the hypothalamus or pituitary gland [16]. In contrast, some authors have observed a higher prevalence of this syndrome in primiparous cows [3,5,12,14], possibly due to a reduced ability to adapt to the metabolic demands of lactation at an early age.

Regarding the risk factors, it should be clarified as to why some of the collected data were not included in the final model. Many of the factors studied are interrelated and may influence outcomes indirectly through others. In this way, twins, stillbirths, and retained placenta are closely associated with dystocia or endometritis, which are included in the model. In the case of subclinical mastitis, it is a risk factor for clinical mastitis, which is present in the model. Furthermore, for primiparous cows, many diseases commonly observed in multiparous cows have a lower incidence in this group.

### 4.1. Reproductive Pathologies

The development of reproductive pathologies during the postpartum period increased the risk of RB occurrence in the animals included in our study. Dystocia was an important risk factor in both primiparous and multiparous cows, aligning with the results described in other studies [3,11,41]. Dystocia has been associated with reduced conception rates and more days open [31], as it delays uterine involution, increases the risk of uterine pathologies [32,42], and delays the resumption of ovarian activity [43]. Additionally, several authors have demonstrated the association between RB and the aforementioned abnormal resumption of ovarian activity [12,16]. In this study, postpartum uterine infections significantly increased the likelihood of RB. In this respect, the presence of endometritis was a risk factor in both primiparous and multiparous cows, consistent with the results reported in several studies [5,7,11,44]. Moreover, particular in multiparous cows, metritis promoted the occurrence of this syndrome, which is in line with the observations of other authors [5,6,11,40,44]. It is known that postpartum uterine pathologies affect cows’ fertility [45,46], as they alter the uterine environment and ovarian function, consequently reducing gamete and embryo survival [33,47,48]. However, previous research conducted in the same geographical area as this study did not report a significant association of dystocia or metritis with higher risk of RB [7].

### 4.2. Specific Animal Factors

Based on the records obtained in this study, a marked BC loss after calving significantly increased the risk of RB development in both primiparous (losses of 0.75 and between 1–1.5 points) and multiparous cows (losses of 1–1.5 points). Without considering BC loss during the transition period, other authors have described a higher prevalence of RB in cows with poor BCS compared to those with moderate or good BCS around parturition [10,11,14,49]. Additionally, previous studies have associated postpartum BC loss with reduced fertility [35,50]; specifically, either a too low BCS at first AI (≤2.50) or significant BC loss in the first three postpartum weeks led to a reduction in the conception rate and reduced embryo quality [51]. These detrimental effects may be due to the negative energy balance that occurs in the peripartum period due to the increased demand for energy associated with the strong increased milk production during the early postpartum period [52]. This negative energy phenomenon can lead to immune function disturbances [53,54], which in turn may increase the risk of uterine pathologies, as well as ovarian reproductive disturbances and lower fertility [55,56,57]. However, other studies have not observed a direct association between BCS and RB incidence [7,12,58].

### 4.3. Metabolic Diseases

Metabolic disorders have also been studied as a potential risk factor for RB. In our study, ketosis significantly increased the risk of RB in both primiparous and multiparous cows. In agreement with these results, Jeong and Kim [7] reported a significant increase in RB incidence in animals that suffer from metabolic disorders like hypocalcemia and ketosis. Additionally, cows with hypocalcemia have been reported to have a higher likelihood of RB [5]; this descent in blood calcium levels may increase dystocia incidence, which has been shown to be also associated with higher RB incidence. Regarding abomasum displacement, which is often associated with ketosis or hypocalcemia, this study did not find a statistically significant effect on RB risk. Conversely, previous studies have suggested an adverse effect of abomasum displacement on fertility in dairy cows [8,9] as well as a higher risk of RB [7].

### 4.4. Other Diseases

In this study, clinical mastitis was a risk factor for RB development in both primiparous and multiparous cows. For the latter, subclinical mastitis was also shown to be a risk factor. These results align with those reported by Befekadu et al. [10], who observed a 2.63 times higher risk of RB in cows that were diagnosed for mastitis. It is important to consider that cases of mastitis may have a direct or indirect effect, potentially indicating poor management practices within the operation, which in turn can negatively impact reproductive performance. Likewise, Gustafsson and Emanuelson [3] described a higher proportion of cows with RB in herds with high clinical mastitis incidence. The negative effects of mastitis on fertility have been previously described, being associated with lower conception rates [30,59] and longer calving-to-conception intervals [30,60,61]. Notably, these detrimental effects on fertility have been associated with both clinical and subclinical mastitis [62]. Cows that suffer from mastitis are thought to experience an increase in proinflammatory cytokine synthesis during infection [63,64,65], which together with bacterial endotoxins causes hormonal imbalances affecting folliculogenesis and ovulation [37,62], as well as early embryo survival [64].

Moreover, lameness also has shown to significantly increase the risk of RB in the animals included in this study, like mastitis mentioned above [66]. Increased services per conception [39] and extended calving-to-conception intervals [67] have been reported in lame cows compared to healthy herd mates. These lameness-associated infectious processes may affect fertility through the aforementioned proinflammatory mechanisms but also do cause pain and stress, resulting in increased glucocorticoid and catecholamine levels in plasma, which in turn alter follicular development and ovulation [39,68].

Data analysis for primiparous cows revealed a significantly lower risk of RB when parturitions studied occurred in summer or autumn. While some authors have not observed a relationship between calving season and RB [12,13], others described a higher RB incidence in winter [49] or autumn–winter compared to summer and autumn [3,69].

It should be noted that the increased incidence during autumn and winter observed in Scandinavia may be due to the influence of a reduced length of the photoperiod during these seasons on hormonal secretion, which could lead to reproductive disturbances [3]. Moreover, a higher incidence of RB during spring and summer has been reported [5], which may be due to the negative effects of heat stress on fertility. These detrimental effects can mainly be explained by a reduced DMI and hence the occurrence of an NEB status, which lead to disruptions in hormone production and follicular development [70,71], as well as in the manifestation of estrus [19,70], impacting the quality of oocytes and hence embryo survival [19,72].

Delaying the first postpartum insemination reduced the risk of RB in multiparous cows in this study. These results indicate that, considering the OR, for each additional day to the voluntary waiting period, the likelihood of RB decreases by 0.01. In this regard, Jeong et al. [7] reported a significantly lower risk of RB in cows inseminated after 80 days postpartum compared to those inseminated ≤ 80 days after calving. Additionally, other studies have previously noted an association between a shorter interval from calving to first insemination and a higher incidence of RB [3,12]. This effect may be due to the fact that longer voluntary waiting periods allow for improved uterine health, a reduction in postpartum related diseases and reduced systemic inflammation, and a sooner reestablishment of ovarian cyclicity [73]. Regarding subclinical endometritis, which has been proven to be one of the major causes of RB, a higher incidence has been reported when shorter calving-to-first-insemination intervals were used [74].

The findings of this study suggest several practical recommendations for reducing RB prevalence. Nutritional management is crucial to prevent excessive body condition loss during the transition period by providing balanced diets tailored to the energy demands of lactating cows. Routine postpartum health monitoring should be conducted to identify and address reproductive and metabolic disorders early, including prompt treatment of mastitis, ketosis, and lameness. Strategic breeding protocols, such as delaying the first insemination postpartum in multiparous cows, can significantly enhance reproductive outcomes. Additionally, optimizing calving periods to align with seasons offering protective benefits, such as summer and autumn as observed in this study, can further reduce the risk of RB.

While this study provides valuable insights, it is essential to acknowledge its limitations. The study’s geographic focus on northern Spain limits the generalizability of findings to other regions with differing management systems, climates, or genetic herds. Farm-specific practices and incomplete historical records may introduce biases in the results, and genetic predispositions and long-term reproductive outcomes were not assessed, presenting an area for future investigation.

Future research should explore RB in diverse geographical and management settings, investigate genetic influences, and evaluate the effectiveness of advanced diagnostic and artificial insemination techniques. A multidisciplinary approach integrating these factors is necessary to enhance reproductive performance further.

## 5. Conclusions

This study identified a prevalence of RB in northern Spain of 21.1% (18.6% in primiparous cows and 22.8% in multiparous cows). Risk factors for RB in primiparous cows included marked body condition loss, ketosis, lameness, clinical mastitis, and reproductive pathologies (dystocia and endometritis), while calving in summer and autumn reduced the likelihood of RB. For multiparous cows, additional factors included metritis and subclinical mastitis, with a protective effect observed when the first postpartum artificial insemination was delayed.

Addressing these factors through targeted management strategies can improve reproductive efficiency, reduce economic losses, and support sustainable dairy farming practices. Future research should focus on addressing the identified limitations, exploring genetic predispositions, and developing advanced tools for diagnosing and managing RB according to the characteristics of the farm.

## Figures and Tables

**Table 1 animals-15-00266-t001:** List of information collected, since calving, from 2370 Holstein cows.

Collected Data	Definition	References
**Reproductive parameters**		
Calving date	Date of the latest calving	
Calving number	Number of calvings	
Number of artificial inseminations needed to conceive after previous calving	Number of artificial inseminations required to achieve pregnancy in the previous calving	
Date of first postpartum insemination	Date of the first insemination performed after the last calving	[30]
**Reproductive diseases**		
Repeat breeder syndrome	Cows that have calved at least once, have not become pregnant after 3 or more inseminations, and show no detectable reproductive abnormalities following an examination of the genital tract using transrectal ultrasound	[4]
Dystocia	Difficult calving that requires human intervention	[31]
Stillbirth	Birth of a full-term calf that is dead	[31]
Twin birth	Birth of two full-term calves	[31]
Retained placenta	Presence of fetal membranes in the genital tract ≥ 24 h after calving	[1]
Metritis	Inflammation of the uterine wall, characterized by an abnormally enlarged uterus, reddish-brown foul-smelling discharge, systemic symptoms, and fever, occurring within the first 21 days after calving	[32]
Endometritis	Inflammatory process of the endometrium, accompanied by purulent or mucopurulent vaginal discharge, without systemic symptoms, occurring 21 days or more after calving	[33]
**Metabolic Diseases**		
Hypocalcemia	Decrease in serum calcium concentration	[34]
Ketosis	Elevated concentration of ketone bodies in the blood	[35]
Displaced abomasum	Distension of the abomasum with an accumulation of fluid and/or gas, displacing it to the left or right and dorsally within the abdominal cavity	[9]
Body condition at 30 days before the date predicted of the calving and 30 days postpartum	Score from 1 (very thin) to 5 (very fat) based on the amount of fat deposits in the rump area	[36]
**Other diseases**		
Clinical mastitis	Inflammation of the mammary gland with visible abnormalities in the milk	[37]
Subclinical mastitis	Somatic cell count > 200,000 cells/mL	[38]
Lameness	Abnormal gait due to limb lesions	[39]
Lung infection	Respiratory tract condition with both local and systemic symptoms	[31]
		[36]

**Table 2 animals-15-00266-t002:** Data collected from farm management software for the multiparous cows (*n* = 1445) included in the study.

Collected Data	Definition
Date of previous calving	Date when the previous calving took place
Calving number	Number of calvings
305-day standardized production	Milk production (kg) standardized from DIM 1 to DIM 305
Dry period duration	Period between the end of one lactation and the start of the next lactation

DIM = days in milk.

**Table 3 animals-15-00266-t003:** Independent variables included in the statistical analysis and corresponding classification of the data collected.

Independent Variable	Data Classification
Calving season	Winter/Spring/Summer/Autumn
Change in BC between dry/pre-calving and postpartum	+0.25 to −0.25/−0.5/−0.75/−1 to −1.5
Number of artificial inseminations required to achieve pregnancy in the previous calving	1/2/3/>3.
Interval between the current calving and the first insemination	Days
Postpartum pathologies (dystocia, twin births, stillbirths, retained placenta, metritis, endometritis)	YES/NO
Metabolic conditions (ketosis and displaced abomasum)	YES/NO
Other conditions (clinical and subclinical mastitis and lameness)	YES/NO

BC = body condition.

**Table 4 animals-15-00266-t004:** Descriptive statistics and odds ratios for the variables included in the logistic regression model analyzing the risk factors for repeat breeder syndrome in primiparous dairy cows.

Variable	Levels	RB/Total (%Cows with RB)	OR (95% CI)	*p*-Value
Calving season	Winter	55/259 (21.2%)	Reference	0.002
	Spring	49/217 (22.6%)	1.084 (0.660–1.302)	0.751
	Summer	27/199 (13.6%)	0.400 (0.223–2.298)	0.002
	Autumn	41/250 (53.6%)	0.529 (0.352–2.249)	0.047
BCS change between calving and 15 D PP	+0.25 to −0.25	17/160 (10.6%)	Reference	<0.001
	−0.5	85/540 (15.7%)	1.672 (0.907–3.080)	0.099
	−0.75	57/195 (29.2%)	2.919 (1.500–5.682)	0.002
	−1 to −1.50	13/28 (16.4%)	8.094 (2.817–23.258)	<0.001
**Reproductive diseases**				
Dystocia	NO	143/870 (16.4%)	Reference	<0.001
	YES	29/55 (52.7%)	5.179 (2.665–10.063)	
Metritis	NO	160/905 (17.7%)	Reference	0.098
	YES	12/20 (60.0%)	2.380 (0.853–6.645)	
Endometritis	NO	146/887 (16.5%)	Reference	<0.001
	YES	26/38 (68.4%)	12.234 (5.548–26.975)	
**Metabolic diseases**				
Ketosis	NO	157/892 (17.6%)	Reference	0.003
	YES	15/33 (45.5%)	3.497 (1.546–7.908)	
**Other diseases**				
Lameness	NO	152/866 (17.6%)	Reference	0.003
	YES	15/33 (45.5%)	3.497 (1.546–7908)	
Clinical mastitis	NO	123/794 (15.5%)	Reference	<0.001
	YES	49/131 (37.4%)	4.221 (2.656–6.709)	

BCS = body condition score; RB: repeat breeder syndrome; OR = odds ratio; CI = confidence interval; PP = postpartum. Winter = January–March. Spring = April–June. Summer = July–September. Autumn = October–December.

**Table 5 animals-15-00266-t005:** Descriptive statistics and odds ratios for the variables included in the logistic regression model analyzing the risk factors for repeat breeder syndrome in multiparous dairy cows.

Variable	Levels	RB/Total (%Cows with RB)	OR (CI)	*p*-Value
Calving season	Winter	101/418 (24.2%)	Reference	0.013
	Spring	79/288 (27.4%)	1.304 (0.882–1.928)	0.183
	Summer	65/356 (18.3%)	0.694 (0.463–1.042)	0.078
	Autumn	80/383 (20.9%)	0.747 (0.507–1.099)	0.139
BCS change between calving and 15 D PP	+0.25 to −0.25	23/158 (14.6%)	Reference	0.022
	−0.5	125/666 (18.8%)	1.035 (0.609–1.759)	0.900
	−0.75	112/436 (25.7%)	1.378 (0.798–2.381)	0.250
	−1 to −1.50	65/180 (36.1%)	1.951 (1.059–3.594)	0.032
Interval calving first AI		581/1445	0.992 (0.987–0.997)	0.002
**Reproductive diseases**				
Dystocia	NO	259/1289 (20.1%)	Reference	0.003
	YES	66/156 (42.3%)	1.905 (1.250–2.902)	
Metritis	NO	288/1382 (20.8%)	Reference	
	YES	37/63 (58.7%)	2.586 (1.377–4.854)	0.003
Endometritis	NO	263/1343 (19.6%)	Reference	<0.001
	YES	62/102 (60.8%)	6.176 (3.434–10.215)	
**Metabolic diseases**				
Abomasum displacement	NO	293/1374 (21.3%)	Reference	
	YES	32/71 (45.1%)	1.722 (0.920–3.224)	0.089
Ketosis	NO	281/1366 (20.6%)	Reference	<0.001
	YES	44/79 (55.7%)	4.285 (2.417–7.597)	
**Other diseases**				
Clinical mastitis	NO	210/1189 (17.7%)	Reference	<0.001
	YES	115/256 (44.9%)	4.154 (2.986–5.780)	
Subclinical mastitis	NO	177/953 (18.6%)	Reference	
	YES	148/492 (30.1%)	1.866 (1.393–2.498)	<0.001
Lameness	NO	275/1303 (21.1%)	Reference	<0.001
	YES	50/142 (35.2%)	2.494 (1.619–3.842)	

BCS = body condition score; RB: repeat breeder syndrome; OR = odds ratio; CI = confidence interval; PP = postpartum; AI: artificial insemination.

## Data Availability

Data will be available from the corresponding author under reasonable request.

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
