# Peer review of "A Prospective Study of the Prevalence and Predictive Risk Factors of Repeat Breeder Syndrome in Dairy Cattle in the North of Spain"

_animals, 2025, doi:10.3390/ani15020266_

Round 1
Reviewer 1 Report
Comments and Suggestions for Authors
Dear authors of manuscript Animals-3377466,
With pleasure an interest I have reviewed your submitted manuscript of which I believe is of interest to publish although the claim the presented data are original and new is not directly true to my sincere opinion. The manuscript is all over well scientifically written and therefore good to understand. Nevertheless, this current version needs major revision before to my opinion it may be accepted for publication. The list of comments to improve the manuscript is provided in the attached file, the comments are depicted directly on the manuscript. Please, run through this list and consider the proposed remarks and suggestions for improvement the manuscript.
Success! REVIEWER

Author Response
We would like to thank all the reviewers for taking their time to revise our manuscript and made the necessary comments and suggestions to improve its quality. In this document, we address all issues rose and give some explanations that hopefully are enough to answer all pending concerns about the manuscript.
Reviewer 1:
Dear authors of manuscript Animals-3377466,
With pleasure an interest I have reviewed your submitted manuscript of which I believe is of interest to publish although the claim the presented data are original and new is not directly true to my sincere opinion. The manuscript is all over well scientifically written and therefore good to understand. Nevertheless, this current version needs major revision before to my opinion it may be accepted for publication. The list of comments to improve the manuscript is provided in the attached file, the comments are depicted directly on the manuscript. Please, run through this list and consider the proposed remarks and suggestions for improvement the manuscript.
one may argue if clinically speaking all anatomic abnormalities can be diagnosed/identified, for example blocked oviducts (which is generally clear at pathology)
The manuscript was modified to clarify the type of examination. Oviductal blockage is difficult to diagnose in vivo, as it requires specific techniques such as the phenolsulphontalein test (PSP test) or contrast ultrasound, according to the studies published to date. In this regard, our group has recently published a research paper about this topic (https://doi.org/10.1016/j.rvsc.2024.105511), observing a low prevalence of this issue in the population studied.
Consider to change this sentence: 'This disorder has been shown to be of........'
The manuscript was modified according to the reviewer’s suggestions.
What do you exactly mean by 'control'?
We mean the management of the disease, regarding both treatment and prevention. The manuscript was modified to clarify this matter.
- heat stress
The manuscript was modified according to the reviewer’s suggestions.
metabolic stress due to NEB during the peri-partal period
The manuscript was modified according to the reviewer’s suggestions.
life time
The manuscript was modified according to the reviewer’s suggestions.
to me this sentence sounds 'cryptic', what do you mean to say here?
The sentence was removed from the manuscript to avoid confusion.
Documented
The manuscript was modified according to the reviewer’s suggestions.
consider Sub-optimal
The manuscript was modified according to the reviewer’s suggestions.
do you mean bunk space or lying (cubical) space or both
The manuscript was modified according to the reviewer’s suggestions.
do cows make at risk to become...
The manuscript was modified according to the reviewer’s suggestions.
reduced DMI impacting reproductive performance
We did not mean to say that just the inadequate feeding would affect reproductive performance, but that all those factors together would increase the risk of RB and this would lead to poor reproductive performance. The manuscript was modified to clarify this.
that have been documented to be
The manuscript was modified according to the reviewer’s suggestions.
and hence economic efficiency.
The manuscript was modified according to the reviewer’s suggestions.
GENERAL remark for INTRODUCTION: watch out to repeat, limit the number of repetitions!!
The manuscript was modified according to the reviewer’s suggestions.
put a value here because it is impossible to understand for 100%; for example: 'to understand them better....' or think of reconstructing this statement sentence
The manuscript was modified according to the reviewer’s suggestions.
significantly/widely
The manuscript was modified according to the reviewer’s suggestions.
add for example: by analyzing repro data collected on dairy farms. What I meant to say: depict how this study was performed, based on what data information or in silico modeling for example
The manuscript was modified according to the reviewer’s suggestions.
To me this Table requests ordering, in this presentation it reads chaotic. In other words: improve the depicted lay out of this Table
All tables were modified to improve their layout.
Clarify this definition, it is not fully clear what you mean here. Ah, clear now on the page 5! Good example that the Table needs lay out improvement :)
All tables were modified to improve their layout.
How were these (historical) data selected that has been used for the calculations.
The selected data included all parameters that were available in the software and that are widely known to influence postpartum health (previous calving, parity, milk production and dry period duration).
You may consider to use DIM (Days in Milk)
The manuscript was modified according to the reviewer’s suggestions.
Add: lactation
The manuscript was modified according to the reviewer’s suggestions.
for a better view, one may consider to depict these independent variable in a Table, instead of to sum those up
The manuscript was modified according to the reviewer’s suggestions.
Rephrase sentence: 'Cases that showed missing or doubtful data were excluded for statistical analysis.
The manuscript was modified according to the reviewer’s suggestions.
Consider to mention ranges (or data per farm)
The manuscript was modified according to the reviewer’s suggestions.
Due to......??
The manuscript was modified according to the reviewer’s suggestions
See comment remark at Table 1, also reorder this Table for clarification and reading
All tables were modified to improve their layout.
check this notation
The manuscript was modified according to the reviewer’s suggestions.
See comments of Table 1 and 3
All tables were modified to improve their layout.
In the present study, the prevalence.....
The manuscript was modified according to the reviewer’s suggestions.
consider 'parity'
The manuscript was modified according to the reviewer’s suggestions.
consider this sentence, not fully clear what you want to depict, what you mean. Is it connected the first part of the sentence, as a consequence, or do you want to put this item forward separately?
The manuscript was modified according to the reviewer’s suggestions to clarify this sentence.
Rewrite this sentence, it is now a bit listed, not a clear stated sentence having a clear message
The manuscript was modified according to the reviewer’s suggestions.
Reduced
The manuscript was modified according to the reviewer’s suggestions.
You do use to contras in this sentence: 'Conversely' and 'contrast'. Reconsider this sentence; moreover, avoid 'some'
The manuscript was modified according to the reviewer’s suggestions.
more open days
The manuscript was modified according to the reviewer’s suggestions.
In this respect, the presence of
The manuscript was modified according to the reviewer’s suggestions.
particular in
The manuscript was modified according to the reviewer’s suggestions.
which is
The manuscript was modified according to the reviewer’s suggestions.
cows'
The manuscript was modified according to the reviewer’s suggestions.
Consequently
The manuscript was modified according to the reviewer’s suggestions.
as this study
The manuscript was modified according to the reviewer’s suggestions.
to be
The manuscript was modified according to the reviewer’s suggestions.
significant or.....
The manuscript was modified according to the reviewer’s suggestions.
clarify this period, be more precise
The manuscript was modified according to the reviewer’s suggestions.
at what time, around/at parturition or...
The manuscript was modified according to the reviewer’s suggestions
or
The manuscript was modified according to the reviewer’s suggestions.
either a too low
The manuscript was modified according to the reviewer’s suggestions.
Reduced
The manuscript was modified according to the reviewer’s suggestions.
the strong increased milk production during the early post partum period (or after calving)
The manuscript was modified according to the reviewer’s suggestions.
energy or DMI or nutrional
The manuscript was modified according to the reviewer’s suggestions.
Phenomenon
The manuscript was modified according to the reviewer’s suggestions.
Ovarian
The manuscript was modified according to the reviewer’s suggestions.
Direct
The manuscript was modified according to the reviewer’s suggestions.
that suffer from metabolic disorders like.....
The manuscript was modified according to the reviewer’s suggestions.
which has been shown to be
The manuscript was modified according to the reviewer’s suggestions.
or hypocalcaemia
The manuscript was modified according to the reviewer’s suggestions.
which is
The manuscript was modified according to the reviewer’s suggestions.
shown to be
The manuscript was modified according to the reviewer’s suggestions.
that were diagnosed for mastitis
The manuscript was modified according to the reviewer’s suggestions.
Remark: this may be a direct but also an indirect effect since a high incidence of clinical mastitis may be interpreted as a signal for sub-optimal (poor) quality of the all over farm management (when you speak about risk factors).
The manuscript was modified according to the reviewer’s suggestions
that suffer from mastitis
The manuscript was modified according to the reviewer’s suggestions.
in turn and together
The manuscript was modified according to the reviewer’s suggestions.
Early
The manuscript was modified according to the reviewer’s suggestions.
also have shown to significantly increase the risk of RB
The manuscript was modified according to the reviewer’s suggestions.
like mastitis as mentioned before,
The manuscript was modified according to the reviewer’s suggestions.
herd mates
The manuscript was modified according to the reviewer’s suggestions.
lameness associated infectious
The manuscript was modified according to the reviewer’s suggestions.
May
The manuscript was modified according to the reviewer’s suggestions.
Do
The manuscript was modified according to the reviewer’s suggestions.
in turn
The manuscript was modified according to the reviewer’s suggestions.
parturitions studied
The manuscript was modified according to the reviewer’s suggestions.
Described
The manuscript was modified according to the reviewer’s suggestions.
compared to summer and autumn
The manuscript was modified according to the reviewer’s suggestions.
length of the
The manuscript was modified according to the reviewer’s suggestions.
A
The manuscript was modified according to the reviewer’s suggestions.
moreover, several
The manuscript was modified according to the reviewer’s suggestions.
mainly explained by a reduced DMI and hence the occurrence of a NEB status
The manuscript was modified according to the reviewer’s suggestions.
quality and
The manuscript was modified according to the reviewer’s suggestions.
and hence
The manuscript was modified according to the reviewer’s suggestions.
consider to skip this sentence, it does not add any extra information
The manuscript was modified according to the reviewer’s suggestions.
Quantify
The manuscript was modified according to the reviewer’s suggestions
in post partum related diseases and hence
The manuscript was modified according to the reviewer’s suggestions.
a sooner
The manuscript was modified according to the reviewer’s suggestions.
proven to be one of the major causes
The manuscript was modified according to the reviewer’s suggestions.
positive
impact and (in)direct consequences
to unravel pathophysiological related factors and to develop............
The conclusion section was entirely modified according to several suggestions.
References in capital

Reviewer 2 Report
Comments and Suggestions for Authors
The authors have presented a valuable study that analyzes the prevalence and predictive risk factors of Repeat Breeding Syndrome (RB) in dairy cows, offering important insights for veterinary clinical practice. The strength of this study lies in its extensive field investigation and practical focus, addressing critical reproductive health issues in dairy cattle. However, the manuscript suffers from issues related to logical coherence in the writing, which require improvement to enhance clarity and fluency. It is also recommended that the authors publish the foundational dataset on an appropriate website or database to ensure transparency and facilitate further validation. In addition, the manuscript has the following issues:
[Simple Summary]
1. This summary is clear and informative.
[Abstract]
2. According to the journal's requirements, "The abstract should be a total of about 200 words maximum." The Abstract can be slightly condensed to comply with this word limit.
3. In the context of epidemiological studies, "OR" (Odds Ratio) is often considered more professional than percentages. However, for improved readability, the use of percentages can be considered. If the authors choose to retain "OR," the full term, "Odds Ratio (OR)," should be introduced upon its first appearance in the manuscript.
4. Line 33: "However, summer (OR: 0.4) or autumn (OR: 0.6) calvings lowered the probability of suffering from this syndrome." This sentence is ambiguous. To clarify, what is the reference point for "lowered the probability"? Is it in comparison to the overall OR or to the OR for spring and winter calvings?
5. Line 41: "a longer calving-to-first-insemination interval in multiparous cows" is ambiguous. What specific duration in days qualifies as "longer"? Defining this interval clearly would enhance precision.
[Introduction]
6. Line 47: The term “repeat breeder syndrome” is unnecessarily bolded and should be formatted consistently with the rest of the text.
7. The first and second paragraphs of the Introduction lack proper transitions and coherence. When the authors mention "multifactorial etiology," they do not proceed to analyze or discuss the specific etiological factors but instead shift abruptly to the "prevalence of RB" in the next paragraph. This structure is confusing for readers and should be improved for logical flow.
8. The purpose of discussing the "prevalence of RB" in the second paragraph is unclear. Based on the authors' writing logic, it would be more appropriate to focus on the low prevalence rates of RB in the United Kingdom and Sweden, analyzing the advantages of their "diverse production systems, management techniques, herd size, and genetic merit." This analysis should not be delayed until the third paragraph, which currently addresses the impact on milk production and the economy.
9. The authors should emphasize the existing gaps or deficiencies in Spain’s current RB prevention and management strategies to highlight the significance of this study.
10. Lines 48–69: This content does not warrant a separate paragraph and should be integrated into the surrounding text for improved flow.
11. Lines 70–71: The authors should explicitly specify the environmental factors being discussed. For instance, providing an example such as "decreases methane emissions per liter of milk produced" would add clarity and depth.
12. The Introduction fails to effectively analyze the causes of RB. The content from Lines 78–86 could be moved to follow Line 53, which would improve the logical structure and ensure a more coherent discussion.
13. Line 77: The authors do not clearly explain how RB affects dairy cow reproduction. Instead, they directly state that "Considering the close relationship between reproductive efficiency and dairy productivity, the identification and treatment of repeat breeders is particularly important." A logical connection explaining the mechanism of RB’s impact on reproduction is missing and needs to be established.
14. Lines 87–88: The Introduction lacks relevant background information on RB in Spain. Simply mentioning the prevalence is insufficient; the authors should describe the impact of RB on the Spanish dairy industry for better contextualization.
15. Overemphasizing economic losses is unnecessary. The authors should carefully consider the logical connections between paragraphs and significantly revise the section to ensure clarity, coherence, and smooth transitions.
[Materials and Methods]
16. Line 102: Please consider explaining how the average (3.1) for the robotic system was calculated, as it contrasts with the range provided for conventional systems. This inconsistency might require clarification or a more standardized presentation to allow for direct comparison.
17. Lines 179–180: The term "automatic milking system with robots" is acceptable; however, some fields may prefer the term "robotic milking system" for conciseness and consistency with common terminology.
18. Please specify how the seasons were categorized (e.g., the months included in each season), as seasonal definitions can vary geographically. For instance, the timing of seasons differs between the northern and southern hemispheres.
19. Lines 135–136: The statement "Affected animals were not included in the corresponding statistical analysis" raises concerns about potential bias, such as exclusion bias, if the missing data is not random. Was the randomness of missing data evaluated (e.g., MAR - Missing at Random)? If not, alternative methods, such as imputation techniques (e.g., multiple imputation), should be considered to avoid a reduction in statistical power and ensure robust analysis.
[Results]
20. The authors conducted extensive and in-depth calculations in the Results section. It is recommended that the raw data be uploaded in Excel format to the MDPI website to facilitate a better understanding of the article’s content for readers.
[Discussion]
21. Lines 174–180: This section mentions that advanced age may affect fertility and provides analysis, but it does not explicitly clarify the association with RB. Are ovarian abnormalities and hormone secretion defects related to RB? If so, this connection should be explicitly stated. If not, the analysis regarding these factors should be removed.
22. Line 196: Was the abbreviation SRC defined earlier in the manuscript? A similar issue arises with AI in Line 203. All abbreviations should be clearly defined when first introduced.
23. Lines 233–238: The logical structure of this paragraph should be adjusted to ensure that the latter part of the analysis is more closely related to RB.
24. To improve the overall logic of the Discussion, it is suggested to group the relevant risk factors into categories for a clearer narrative flow.
25. Summarize and highlight practical management recommendations for RB prevention and control to enhance the applied value of the findings.
26. Include a section discussing the study’s limitations and provide suggestions for future research directions.
[Conclusions]
27. Lines 271–273: The study’s limitations should be presented in detail as a separate paragraph within the Discussion section, rather than being briefly mentioned in the Conclusions.
28. The Conclusions section can be moderately condensed. Additionally, the authors should ensure that this section provides a concise synthesis of the results and emphasizes the ultimate objectives of the study.
[References]
29. As a research article, the number of references is excessive, and it is recommended to streamline them.
Author Response
We would like to thank the reviewer for their valuable feedback and for taking the time to review our manuscript.
Reviewer 2:
The authors have presented a valuable study that analyzes the prevalence and predictive risk factors of Repeat Breeding Syndrome (RB) in dairy cows, offering important insights for veterinary clinical practice. The strength of this study lies in its extensive field investigation and practical focus, addressing critical reproductive health issues in dairy cattle. However, the manuscript suffers from issues related to logical coherence in the writing, which require improvement to enhance clarity and fluency. It is also recommended that the authors publish the foundational dataset on an appropriate website or database to ensure transparency and facilitate further validation. In addition, the manuscript has the following issues:
[Simple Summary]
- This summary is clear and informative.
[Abstract]
- According to the journal's requirements, "The abstract should be a total of about 200 words maximum." The Abstract can be slightly condensed to comply with this word limit.
The abstract was modified according to the reviewer´s suggestions.
- In the context of epidemiological studies, "OR" (Odds Ratio) is often considered more professional than percentages. However, for improved readability, the use of percentages can be considered. If the authors choose to retain "OR," the full term, "Odds Ratio (OR)," should be introduced upon its first appearance in the manuscript.
The abstract was modified according to the reviewer’s suggestions.
- Line 33: "However, summer (OR: 0.4) or autumn (OR: 0.6) calvings lowered the probability of suffering from this syndrome." This sentence is ambiguous. To clarify, what is the reference point for "lowered the probability"? Is it in comparison to the overall OR or to the OR for spring and winter calvings?
Additional information was added to clarify this matter. It was in comparison to calvings in winter/spring.
- Line 41: "a longer calving-to-first-insemination interval in multiparous cows" is ambiguous. What specific duration in days qualifies as "longer"? Defining this interval clearly would enhance precision.
The manuscript was modified according to the reviewer’s suggestions. [Introduction]
- Line 47: The term “repeat breeder syndrome” is unnecessarily bolded and should be formatted consistently with the rest of the text.
The manuscript was modified according to the reviewer’s suggestions.
- The first and second paragraphs of the Introduction lack proper transitions and coherence. When the authors mention "multifactorial etiology," they do not proceed to analyze or discuss the specific etiological factors but instead shift abruptly to the "prevalence of RB" in the next paragraph. This structure is confusing for readers and should be improved for logical flow.
The updated structure aims to enhance its clarity and logical flow, aligning better with the overall narrative of the manuscript. We have ensured that the revised section adheres to your suggestions and provides a more cohesive and comprehensible presentation of the content.
- The purpose of discussing the "prevalence of RB" in the second paragraph is unclear. Based on the authors' writing logic, it would be more appropriate to focus on the low prevalence rates of RB in the United Kingdom and Sweden, analyzing the advantages of their "diverse production systems, management techniques, herd size, and genetic merit." This analysis should not be delayed until the third paragraph, which currently addresses the impact on milk production and the economy.
The manuscript was modified according to the reviewer’s suggestions.
- The authors should emphasize the existing gaps or deficiencies in Spain’s current RB prevention and management strategies to highlight the significance of this study.
The manuscript was modified according to the reviewer’s suggestions.
- Lines 48–69: This content does not warrant a separate paragraph and should be integrated into the surrounding text for improved flow.
The manuscript was modified according to the reviewer’s suggestions.
- Lines 70–71: The authors should explicitly specify the environmental factors being discussed. For instance, providing an example such as "decreases methane emissions per liter of milk produced" would add clarity and depth.
- The Introduction fails to effectively analyze the causes of RB. The content from Lines 78–86 could be moved to follow Line 53, which would improve the logical structure and ensure a more coherent discussion.
- Line 77: The authors do not clearly explain how RB affects dairy cow reproduction. Instead, they directly state that "Considering the close relationship between reproductive efficiency and dairy productivity, the identification and treatment of repeat breeders is particularly important." A logical connection explaining the mechanism of RB’s impact on reproduction is missing and needs to be established.
- Lines 87–88: The Introduction lacks relevant background information on RB in Spain. Simply mentioning the prevalence is insufficient; the authors should describe the impact of RB on the Spanish dairy industry for better contextualization.
The manuscript was modified according to the reviewer’s suggestions.
- Overemphasizing economic losses is unnecessary. The authors should carefully consider the logical connections between paragraphs and significantly revise the section to ensure clarity, coherence, and smooth transitions.
The manuscript was modified according to the reviewer’s suggestions.
[Materials and Methods]
- Line 102: Please consider explaining how the average (3.1) for the robotic system was calculated, as it contrasts with the range provided for conventional systems. This inconsistency might require clarification or a more standardized presentation to allow for direct comparison.
The manuscript was modified according to the reviewer’s suggestions. We reflected in the text that the average used was the one given by default in the robot.
- Lines 179–180: The term "automatic milking system with robots" is acceptable; however, some fields may prefer the term "robotic milking system" for conciseness and consistency with common terminology.
The manuscript was modified according to the reviewer’s suggestions.
- Please specify how the seasons were categorized (e.g., the months included in each season), as seasonal definitions can vary geographically. For instance, the timing of seasons differs between the northern and southern hemispheres.
The manuscript was modified according to the reviewer’s suggestions as a comment in table 4 .
- Lines 135–136: The statement "Affected animals were not included in the corresponding statistical analysis" raises concerns about potential bias, such as exclusion bias, if the missing data is not random. Was the randomness of missing data evaluated (e.g., MAR - Missing at Random)? If not, alternative methods, such as imputation techniques (e.g., multiple imputation), should be considered to avoid a reduction in statistical power and ensure robust analysis.
Due to lack of data in multiparous cows were not included only 61 meanwhile in primiparous cows only 7. The data supporting this study are available upon request from the corresponding author, and we would be delighted to share them with anyone interested.
[Results]
- The authors conducted extensive and in-depth calculations in the Results section. It is recommended that the raw data be uploaded in Excel format to the MDPI website to facilitate a better understanding of the article’s content for readers.
The data supporting this study are available upon request from the corresponding author, and we would be delighted to share them with anyone interested.
[Discussion]
- Lines 174–180: This section mentions that advanced age may affect fertility and provides analysis, but it does not explicitly clarify the association with RB. Are ovarian abnormalities and hormone secretion defects related to RB? If so, this connection should be explicitly stated. If not, the analysis regarding these factors should be removed.
The manuscript was modified according to the reviewer’s suggestions.
- Line 196: Was the abbreviation SRC defined earlier in the manuscript? A similar issue arises with AI in Line 203. All abbreviations should be clearly defined when first introduced.
The manuscript was modified according to the reviewer’s suggestions.
- Lines 233–238: The logical structure of this paragraph should be adjusted to ensure that the latter part of the analysis is more closely related to RB.
The manuscript was modified according to the reviewer’s suggestions.
- To improve the overall logic of the Discussion, it is suggested to group the relevant risk factors into categories for a clearer narrative flow.
The manuscript was modified according to the reviewer’s suggestions.
- Summarize and highlight practical management recommendations for RB prevention and control to enhance the applied value of the findings.
The manuscript was modified according to the reviewer’s suggestions.
- Include a section discussing the study’s limitations and provide suggestions for future research directions.
The manuscript was modified according to the reviewer’s suggestions.
[Conclusions]
- Lines 271–273: The study’s limitations should be presented in detail as a separate paragraph within the Discussion section, rather than being briefly mentioned in the Conclusions.
The manuscript was modified according to the reviewer’s suggestions.
- The Conclusions section can be moderately condensed. Additionally, the authors should ensure that this section provides a concise synthesis of the results and emphasizes the ultimate objectives of the study.
The manuscript was modified according to the reviewer’s suggestions.
[References]
- As a research article, the number of references is excessive, and it is recommended to streamline them.
We understand the importance of maintaining a concise and focused reference list. To address this, we have already made significant efforts to condense the references by prioritizing the most relevant and recent works that directly support the key points of our study. The current reference list reflects what we consider essential to provide a comprehensive context and foundation for our findings, while also acknowledging the contributions of previous studies in the field.

Reviewer 3 Report
Comments and Suggestions for Authors
Brief summary:
Authors designed to determine the risk factors involved in the development of repeat breeder (RB) syndrome in dairy cattle farms in northern Spain.
They concluded that the risk factors of RB for multiparous cows were the same as those for primiparous cows, except with metritis and subclinical mastitis and that understanding the etiology and pathophysiology of RB remains incomplete.
Broad comments:
This study analyzed the associations that may lead to RP in four dairy farms in the north region of Spain. Authors used appropriate statistical methods. The interpretation of the data is understandable, however, association with R2 lower than 0.2 can be considered week. Discussion of the gained relusts is fair, but selection of the used references to limit to the most relevant ones is suggested. The language used is proper and understandable.
Specific comments:
L29: ...condition postpartum…
L102-4: what did include: ...weekly reproductive examinations and uterine health checks during the fourth week postpartum?
L108: In Table 1, the original definitions of metritis and endometritis are classicaly cited from the article by Sheldon et al., 2006 (Theriogenology. 2006 May;65(8):1516-30., doi: 10.1016/j.theriogenology.2005.08.021.)
L141-2 and L155-6: Models with an R² of 0.17 or 0.18 do not represent close associations in primiparous or multiparous dairy cows.
L179: ...age, in contrary to...
L196: Define SRC
References: Authors should check the entire list and accurately follow the Instructions of the Journal. Lots of mistakes and incorrectly or incompletely listed references are affected. A few local and in their content marginal references from non peer-reviewed sources should better be removed.
Author Response
We would like to thank the reviewer for their valuable feedback and taking their time to revise our manuscritp.
Reviewer 3
Brief summary:
Authors designed to determine the risk factors involved in the development of repeat breeder (RB) syndrome in dairy cattle farms in northern Spain.
They concluded that the risk factors of RB for multiparous cows were the same as those for primiparous cows, except with metritis and subclinical mastitis and that understanding the etiology and pathophysiology of RB remains incomplete.
Broad comments:
This study analyzed the associations that may lead to RP in four dairy farms in the north region of Spain. Authors used appropriate statistical methods. The interpretation of the data is understandable, however, association with R2 lower than 0.2 can be considered week. Discussion of the gained relusts is fair, but selection of the used references to limit to the most relevant ones is suggested. The language used is proper and understandable.
Specific comments:
L29: ...condition postpartum…
The manuscript was modified according to the reviewer’s suggestions.
L102-4: what did include: ...weekly reproductive examinations and uterine health checks during the fourth week postpartum?
The reproduction veterinarian visited the farms weekly, and during these visits, he/she examined the animals in the 4th week postpartum, performing a complete reproductive examination, detecting and managing reproductive problems, monitoring their general health status, and collecting the information recorded by the farmer since the last visit.
L108: In Table 1, the original definitions of metritis and endometritis are classicaly cited from the article by Sheldon et al., 2006 (Theriogenology. 2006 May;65(8):1516-30., doi: 10.1016/j.theriogenology.2005.08.021.)
Due to the necessity of condensing the number of references in our manuscript, we made the decision to replace the original citation with another reference that was already included in the text and provides a similar definition. This decision was made to balance the need for brevity with the relevance of the references cited.
L141-2 and L155-6: Models with an R² of 0.17 or 0.18 do not represent close associations in primiparous or multiparous dairy cows.
We acknowledge that the R² value is not very high; however, given the large sample size, the significance of the model, and the non-significant Hosmer-Lemeshow test, we can state with some confidence that our model explains 17%–18% of the variance in the studied population. Considering the importance and complexity of this issue, this percentage is highly valuable for disease prevention. Furthermore, when combined with improvements in diagnosis and treatment, the economic losses caused by SRS would be significantly reduced.
L179: ...age, in contrary to..
The manuscript was modified according to the reviewer’s suggestions.
L196: Define SRC
The manuscript was modified according to the reviewer’s suggestions.
References: Authors should check the entire list and accurately follow the Instructions of the Journal. Lots of mistakes and incorrectly or incompletely listed references are affected. A few local and in their content marginal references from non peer-reviewed sources should better be removed.
We have reviewed the entire reference list and ensured that it now accurately adheres to the Journal’s Instructions for Authors. Any formatting mistakes or incomplete references have been corrected.
